# Integrated Analysis of Differential Expression Profiles of miRNA and mRNA in Gonads of *Scatophagus argus* Provides New Insights into Sexually Biased Gene Expression

**DOI:** 10.3390/ani15111564

**Published:** 2025-05-27

**Authors:** Yaling Lei, Kaizhi Jiao, Yuanqing Huang, Yuwei Wu, Gang Shi, Hongjuan Shi, Huapu Chen, Siping Deng, Guangli Li, Wenjing Tao, Dongneng Jiang

**Affiliations:** 1Guangdong Research Center on Reproductive Control and Breeding Technology of Indigenous Valuable Fish Species, Fisheries College, Guangdong Ocean University, Zhanjiang 524088, China; leiyaling2022@163.com (Y.L.); jiaokaizhi@163.com (K.J.); 11112201005@stu.gdou.edu.cn (Y.H.); wuyuwei9930@163.com (Y.W.); shign@126.com (G.S.); shihongjuan1990@163.com (H.S.); chenhp@gdou.edu.cn (H.C.); dengsp@gdou.edu.cn (S.D.); ligl@gdou.edu.cn (G.L.); 2Key Laboratory of Marine Ecology and Aquaculture Environment of Zhanjiang, Fisheries College, Guangdong Ocean University, Zhanjiang 524088, China; 3Guangdong Provincial Key Laboratory of Aquatic Animal Disease Control and Healthy Culture, Fisheries College, Guangdong Ocean University, Zhanjiang 524088, China; 4Key Laboratory of Freshwater Fish Reproduction and Development (Ministry of Education), Southwest University, Chongqing 400715, China; 5Key Laboratory of Aquatic Science of Chongqing of Life Sciences, Southwest University, Chongqing 400715, China

**Keywords:** *Scatophagus argus*, microRNAs, gonad, sex determination and differentiation, control network

## Abstract

*Scatophagus argus* (*S. argus*) is a key aquaculture species in southern China, with females growing faster than males. Limited knowledge of its sex determination and differentiation hinders sex-controlled breeding. MicroRNAs (miRNAs) regulate these processes in vertebrates, but no research exists on their role in *S. argus*. This study analyzed miRNA/mRNA expression in *S. argus* gonads, identifying 2210 miRNAs (482 sex-differentially expressed) targeting 3340 genes to form 13,773 regulatory pairs. Key sex-related genes (*Foxl2*, *Gdf9*, *Gsdf*, *Sox3*) showed coordinated or inverse expression patterns with their regulatory miRNAs. The species-specific miRNA and gene regulatory network revealed non-conserved mechanisms across fish. The present study advances the understanding of sexual dimorphism and provides potential targets for sex-controlled breeding in this important marine aquaculture species.

## 1. Introduction

The mechanisms of sex determination (SD) in fish can be broadly categorized into three main types: genetic sex determination (GSD), environmental sex determination (ESD), and a mixed mode involving the synergistic regulation of both genetic and environmental factors [1,2]. In GSD systems, key sex-determining genes located on sex chromosomes (such as *Amhy* and *DMY*/*Dmrt1bY*) or polygenic factors on autosomes regulate gonadal development through hierarchical regulatory networks [3,4,5]. These master genes drive sexual differentiation by activating sex-specific signaling pathways (e.g., TGF-β promoting testis development and Wnt/β-catenin facilitating ovarian formation) while modulating steroid hormone biosynthesis [6,7]. Extensive research has revealed an intricate bidirectional regulatory network where sex-determining genes and differentiation genes maintain developmental homeostasis through mutual antagonism: male pathway genes (*Dmrt1*, *Amh*, etc.) suppress female pathway genes (*Foxl2*, *Cyp19a1a*, etc.), and vice versa [2,8]. A well-documented example is found in *Oreochromis niloticus* (*O. niloticus*), where the *Amhy* gene initiates male development by upregulating *Dmrt1* while simultaneously repressing *Foxl2* and *Cyp19a1a* expression, ultimately activating *Gsdf* to drive testicular differentiation [9,10,11]. While substantial progress has identified core regulatory components, our understanding remains incomplete regarding their precise regulatory networks. Particularly, the epigenetic control mechanisms governing these processes (encompassing dynamic DNA methylation, histone modifications, and non-coding RNA regulation) are still in the early stages of investigation, representing a critical frontier for future research in this field.

Non-coding RNAs (ncRNAs), an important epigenetic modification, have the ability to regulate post-transcriptional mechanisms that influence sex-related gene expression [12,13,14]. MicroRNAs (miRNAs) are endogenous ncRNAs with a length of approximately 22 nucleotides (nt). MiRNAs primarily regulate gene expression by binding to the 3′ untranslated region (UTR) of target mRNAs. In animals, this binding typically leads to translational repression and/or mRNA degradation (via deadenylation) [15,16,17]. These interactions generally establish a negative regulatory relationship between miRNAs and their target mRNA. MiRNAs exert post-transcriptional regulatory functions in both plant and animal species [18]. Deletion of miR-17~92 in mice causes sex reversal, highlighting the key role of miRNAs in sex determination [19]. In *Bactrocera dorsalis* (*B. dorsalis*), miR-1-3p is essential for male determination during early embryogenesis, serving as an intermediate factor in the male sex-determination pathway with *Bdtra* (the female sex-determining gene) as its target. When miR-1-3p is knocked out, the expression of *Bdtra* and the female-specific splice variant of *doublesex* (*Bddsx*) is upregulated, inducing sexual reversal in XY individuals to phenotypic females [20]. Emerging evidence highlights the critical regulatory role of miRNAs in fish sex determination and differentiation through targeted gene modulation. In *Epinephelus coioides* (*E. coioides*), miR-26a-5p directly targets and suppresses *Cyp19a1a* expression, establishing a regulatory axis where estrogen (E2) treatment downregulates miR-26a-5p to promote *Cyp19a1a* expression, thereby facilitating female sex reversal in males [21]. Similarly, in *Monopterus albus* (*M. albus*), miR-430 coordinately regulates multiple key genes (*Cyp19a1b*, *Cyp17*, and *Foxl2*) involved in steroidogenesis and sexual differentiation [22]. Another notable example in *Cynoglossus semilaevis* (*C. semilaevis*) demonstrates that miR-196a-4 inhibits *Lgr8* expression, significantly impacting testicular development [23]. While these findings reveal species-specific miRNA-mediated regulatory mechanisms in sex development, the evolutionary conservation of these miRNA and gene interactions across fish species requires further investigation. Elucidating these miRNA regulatory networks will significantly advance our understanding of vertebrate sexual development mechanisms.

The rapid development of sequencing technology and bioinformatics has paved the way for a new approach to the joint analysis of multi-omics, providing a novel perspective on the regulation of gene expression in relation to other studies [24,25]. Recent studies highlight the crucial role of miRNA–mRNA regulatory networks in fish reproduction: In *Salmo salar* (*S. salar*), stage-specific miRNA–mRNA networks dynamically regulate spermatogenesis, revealing post-transcriptional control of sperm development [26]. In *Gobiocypris rarus* (*G. rarus*), 17α-methyltestosterone (17α-MT)-induced sex reversal involves negatively correlated miRNA–mRNA pairs that modulate steroid biosynthesis pathways [27]. Remarkably, *Oryzias latipes* (*O. latipes*) embryos exhibit adaptive miRNA-mRNA network restructuring in response to environmental changes, highlighting their regulatory plasticity [28]. These findings establish miRNA-mediated post-transcriptional regulation as a key mechanism in gonadal development, sex determination, and environmental adaptation, offering insights for aquaculture improvement.

*Scatophagus argus* (*S. argus*) is an economically important aquaculture and ornamental species in southern China due to its high nutritional value and colorful appearance [29,30]. *S. argus* has an XX–XY sex determination system. It exhibits sexual growth dimorphism, with females growing faster than males, so breeding all-female populations would be beneficial for economic purposes [31,32]. To establish an all-female breeding system in *S. argus*, the successful production of sex-reversed XX males for crossing with normal XX females is crucial for generating all-female progeny. However, conventional hormonal treatments using androgen 17α-MT and aromatase inhibitor letrozole (Le) fail to induce complete sex reversal in XX individuals [31]. This limitation may be attributed to the absence of functional *Dmrt1* (whereas the Y chromosome encodes intact *Dmrt1Y*, the X chromosome carries a truncated *Dmrt1*Δ*X* variant) [32]. The inability to produce viable XX neo-males currently represents a major technical constraint in developing sex-control techniques for *S. argus* aquaculture. The technique of sex control in *S. argus* faces bottlenecks. Therefore, more research should be conducted in the field of sex determination and differentiation to address the challenges of sex-controlled breeding in the *S. argus*, and understanding the expression and regulation of miRNA on sex-related genes may help to solve this problem in *S. argus*. This study predicted the target regulatory relationship between gonadal differentially expressed miRNAs (DEMs) and differentially expressed genes (DEGs) in *S. argus*, and identified its miRNAs that might regulate key sex determination and differentiation genes. It deepens our understanding of the molecular regulatory mechanism of sex determination and differentiation in *S. argus*, and provides a theoretical basis for further studies on its sex-control breeding.

## 2. Materials and Methods

### 2.1. Experimental Animals

The experimental fish, healthy adult *S. argus*, were procured from Dongfeng Aquatic Market in Zhanjiang, Guangdong Province, China. A total of six specimens, comprising three females (designated as F1, F2, and F3) and three males (designated as M1, M2, and M3), were selected for this study. The fish were anesthetized with 100 mg/L 4-allyl-2-methoxyphenol (E809010, Macklin, Shanghai, China) prior to sampling. After completing body weight and length measurements (Appendix A), gonadal tissue samples were immediately collected through dissection. For RNA samples: in females, the gonadal capsule was removed and the central parenchymal tissue was collected (50–100 mg/tube); in males, the entire gonadal tissue including the capsule was collected. All RNA samples were flash-frozen in liquid nitrogen and stored at −80 °C. For histological samples: gonadal tissues with intact capsules were cut into standardized 5 × 5 × 3 mm cubic blocks, fixed in Bouin’s solution for 24 h, followed by paraffin embedding and sectioning. Sex and gonadal stage were determined by hematoxylin and eosin staining using established histological methods [33]. The samples used in this experiment are the same as those were used for the integrated analysis of gonadal methylome and transcriptome by Jiao et al. (2024) [34]. Both male and female gonads are in stage IV according to histological observation [34]. No endangered or protected species were used in this study. All animal experiments were conducted in accordance with the guidelines of the Animal Research and Ethics Committee of Guangdong Ocean University (201903004) (Appendix A).

### 2.2. Total RNA Extraction

Total RNA was extracted from gonadal tissues using TRIzol reagent (Invitrogen, Carlsbad, CA, USA) according to the following protocol: The tissues were homogenized in 1 mL of TRIzol, followed by the addition of 0.2 mL chloroform and centrifugation for phase separation. The aqueous phase containing RNA was precipitated with isopropanol, washed with 75% ethanol, and finally dissolved in RNase-free water. RNA quality was assessed by measuring concentration (Nanodrop 2000, Thermo Scientific, Wilmington, DE, USA), integrity (Agilent 2100 Bioanalyzer, Agilent Technologies, Santa Clara, CA, USA), and purity (LabChip GX, PerkinElmer, Waltham, MA, USA). Only high-quality RNA samples meeting the following criteria were used for subsequent Illumina sequencing and miRNA analysis: total quantity ≥ 1.5 μg (sufficient for two library constructions), concentration ≥ 200 ng/μL, volume ≥ 10 μL, with OD260/280 ratio of 1.7–2.5, OD260/230 ratio of 0.5–2.5, normal 260 nm absorption peak, RIN value ≥ 8, and 28S/18S ratio ≥ 1.0 (preferably ≥ 2.0) with flat or minimally elevated baseline.

### 2.3. Library Preparation and Small RNA (sRNA) Sequencing

Six small RNA libraries were constructed, comprising three biological replicates for each sex, using the VAHTSTM Small RNA Library Prep Kit (Illumina, San Diego, CA, USA).The procedure included: (1) Sequential ligation of 3′ and 5′ adapters (3′NEXTflex Adenylated Adapter and 5′SR Adaptor) at 28 °C for 1 h each; (2) Reverse transcription using M-MuLV Reverse Transcriptase (RNase H-) at 42 °C for 60 min; (3) PCR amplification with LongAmp Taq 2× Master Mix and Illumina-specific primers (15 cycles: 94 °C/15 s, 65 °C/15 s, 72 °C/15 s); (4) Size selection (138–150 bp) on 6% TBE polyacrylamide gels (Thermo Fisher Scientific, USA); (5) Purification using VAHTSTM DNA Clean Beads (Vazyme, Nanjing, China); (6) Quality control on Qsep-400 system (BiOptic Inc., Taipei, China; concentration > 1 ng/μL, no adapter dimers). The qualified libraries were sequenced on Illumina NovaSeq 6000 (PE150; Biomarker Technologies Co., Ltd., Beijing, China), with raw data deposited in NCBI SRA (PRJNA1149578).

### 2.4. miRNAs Identification

Raw sequencing data were preprocessed using fastp [35] to generate clean reads through the removal of adapter contamination, poly-N sequences (>10%), and low-quality bases (Phred score < 20). The resulting high-quality reads, ranging from 18–30 nt in length, were subsequently evaluated for standard quality parameters including Q20, Q30 scores, GC content, and sequence duplication levels. All subsequent analyses were based on the basis of high-quality clean reads. Clean reads were aligned against the Silva (http://www.arb-silva.de/, accessed on 4 March 2024), GtRNAdb (https://lowelab.ucsc.edu/GtRNAdb/, accessed on 4 March 2024), Rfam (http://rfam.xfam.org/, accessed on 4 March 2024), and Repbase (https://www.girinst.org/repbase/, accessed on 4 March 2024) databases using Bowtie (v1.0.0) [36] to filter out non-coding RNAs (ncRNAs) and repetitive sequences, including ribosomal RNAs (rRNAs), transfer RNAs (tRNAs), small nuclear RNAs (snRNAs), and small nuclear ribonucleoproteins (snoRNAs). Unannotated reads containing miRNA were obtained. The unannotated reads were aligned to the *S. argus* reference genome (accession number: GWHAOSK00000000.1; available at https://bigd.big.ac.cn/gwh/, accessed on 4 March 2024) using Bowtie (version 1.0.0), yielding mapped reads with genomic position information (miRNA initial position information). Subsequently, the successfully mapped reads were analyzed against the miRBase database (v22) (http://www.mirbase.org/, accessed on 25 March 2024) to identify conserved miRNAs through sequence homology comparison with known mature miRNA sequences. The identification of known miRNAs was conducted by permitting up to one mismatch in the upstream 2-nt and downstream 5-nt regions during sequence alignment Strict matching of seed regions ensures specificity of target identification; high tolerance for mismatches at both ends is to accommodate sequencing errors or inter-species microvariation. To identify novel miRNAs from remaining mapped reads that do not align to known miRNAs in miRBase database, we performed precursor-based prediction through the following workflow: First, we extracted genomic regions containing these mapped reads and extended them by 200 bp upstream and 50 bp downstream to capture potential precursor hairpin structures and Dicer cleavage sites. These extended sequences were then analyzed using miRDeep2 (v2.0.5) [37]: (1) The Mapper module aligned the remaining mapped reads to the extended sequences to determine their precise locations, abundance, and mismatch patterns (thereby defining pre-miRNA boundaries); (2) The Quantifier module quantified mapped reads on precursors and evaluated candidate reliability using randfold software (v2.0) scores. High-scoring precursors meeting miRNA biogenesis criteria were identified as novel miRNA precursors, with their reads classified as novel miRNAs.

All miRNAs identified from the gonads of *S. argus*, including both known and novel miRNAs, were comprehensively analyzed. The analysis encompassed length distribution, base deviation at the first nucleotide position, nucleotide bias at each position, and the identification of miRNAs undergoing base editing using isomiRID software (v1.0) [38]. Additionally, miRNA family analysis was conducted for both known and novel miRNAs using miRDeep2 (v2.0.5). This analysis required perfect matching of the seed sequence (2–8 nt) to classify miRNAs as belonging to the same family, leveraging sequence similarity for accurate annotation and classification.

### 2.5. MiRNA Expression and Differential Expression Analysis

The distribution and proportion of reads corresponding to identified miRNAs in the gonads were statistically analyzed following normalization of miRNA expression levels using the Trusted Platform Module (TPM) algorithm [39]. DEMs were identified using DESeq2 software (v1.6.3) [40], with miRNAs meeting the criteria of |log2(FC)| > 2 and a corrected *p*-value < 0.01 classified as DEMs.

### 2.6. Verification of miRNA Expression by Real-Time Quantitative PCR (RT-qPCR)

To validate the DEMs identified in this study, the expression levels of randomly selected miRNAs were quantified using RT-qPCR in both ovarian and testicular tissues. A total of eight DEMs were selected for verification, including four miRNAs that were significantly upregulated in the ovary and three miRNAs that exhibited significant upregulation in the testis. Quantification was performed using the poly(A)-tailing-based RT-qPCR method. This approach employed miRNA-specific forward primers designed against mature sequences and a universal reverse primer (R: GATCGCCCTTCTACGTCGTAT) complementary to the 3′-adapter sequence [41]. The forward primers were designed using the online primer design system from Sangon Biotech (Shanghai, China) (https://store.sangon.com/newPrimerDesign, accessed on 7 June 2024). For RT-qPCR analysis, cDNA was synthesized using the TransScript^®^ miRNA First-Strand cDNA Synthesis SuperMix (TransGen Biotech, Beijing, China). Subsequently, the synthesized cDNA was utilized as a template for RT-qPCR amplification with the PerfectStart^®^ Green qPCR SuperMix (TransGen Biotech, China). The RT-qPCR conditions were set as follows: initial denaturation at 95 °C for 3 min, followed by 40 cycles of denaturation at 95 °C for 30 s, annealing at 58 °C for 30 s, and extension at 72 °C for 15 s. The relative expression levels of miRNAs were quantified using the 2^−ΔΔCt^ method, with U6 serving as the internal reference gene. Results are presented as the mean ± standard deviation of triplicate measurements. Statistical comparisons of miRNA expression levels between testis and ovaries were performed using an independent samples *t*-test, with a significance threshold of *p* < 0.05. All primers were synthesized by Sangon Biotech Co., Ltd. (Shanghai, China). Detailed sequence information is provided in Table 1.

### 2.7. Target Gene Prediction

To identify target DEGs regulated by DEMs, we performed differential expression analysis on transcriptome data from the same samples (PRJNA906196 and PRJNA1030442) [34], using the same reference genome as the miRNA analysis (GWHAOSK00000000.1; https://bigd.big.ac.cn/gwh/, accessed on 4 March 2021). Significant DEGs were identified with thresholds of |log2FC| > 2 and FDR < 0.01. Given that miRNA-mediated gene regulation primarily occurs through sequence complementarity with 3′UTRs, we extracted annotated 3′UTR sequences from the reference genome using TBtools-ll (v2.025). For genes lacking 3′UTR annotations, we developed a Python (v3.11.4) pipeline to obtain these regions, defined as sequences between stop codons and polyadenylation signals. This generated a comprehensive DEG-specific 3′UTR dataset for subsequent analysis. The binding sites of DEMs to the 3′UTRs of DEGs were predicted based on the complementarity between miRNAs and the 3′UTRs, as well as the free energy of RNA–RNA duplexes. Predictions were performed using miRanda (v3.3a) [42] and TargetScan (v5.2) [43] with specific parameters. For miRanda, the minimum free energy (S) threshold was set at ≥150 kcal/mol, and the Gibbs free energy change (ΔG) threshold was ≤−25 kcal/mol. For TargetScan, target prediction is primarily based on complementarity between the miRNA 5′ end seed region (2–8 nt) and the 3′UTR of target genes.

### 2.8. GO and KEGG Enrichment Analysis

After establishing the targeting relationships between DEMs and DEGs, the DEGs containing binding sites for DEMs were functionally analyzed using gene annotation data from the Gene Ontology (GO) and Kyoto Encyclopedia of Genes and Genomes (KEGG) databases. This analysis aimed to identify additional functional categories associated with the candidate target genes. Significantly enriched terms and pathways were defined as those with a corrected *p* < 0.05 in the GO and KEGG enrichment analyses.

## 3. Results

### 3.1. Construction of miRNA Library

Six small RNA libraries were constructed and sequenced. After sequencing, a total of 2.4 Gb of raw sequencing data was obtained from six small RNA libraries, with an average Q30 value exceeding 95.93% across all libraries. After the removal of contaminants and poor-quality sequences, a total of 76.60 Mb of clean reads was obtained, with more than 9.88 Mb of clean reads for each sample (Table 2).

### 3.2. Identification and Classification of miRNAs

Through comparison with the miRBase database and analysis of miRNA biosignatures, a total of 2210 miRNAs were identified, comprising 779 known miRNAs and 1431 novel miRNA candidates (Appendix A). Among these miRNAs, 1161 miRNAs were annotated and classified into 218 families, and the ten most abundant miRNA families were let-7, mir-133, mir-9, mir-17, mir-27, mir-455, mir-140, mir-10, mir-30, and mir-153 (Figure 1).

TPM normalization analysis revealed that the majority of miRNAs (37.15%, *n* = 821) exhibited expression levels from 10–100, followed by those in the 0–10 range (34.84%, *n* = 770). miRNAs with expression >1000 accounted for the smallest proportion (5.88%, *n* = 130) (Figure 2A). In addition, the proportion of miRNAs with different ranges of expression levels was similar in the ovary and testis (Figure 2B). In detail, the majority of known miRNAs had length 22 nt, whereas newly predicted miRNAs showed a peak at 25 nt (Figure 2C). Nucleotide composition analysis revealed that a consistent uracil (U) bias in both known and novel miRNAs, accounting for 30.66% and 30.33%, respectively. While conserved miRNAs exhibited a nucleotide preference order of U > adenine (A) > guanine (G) > cytosine (C), novel miRNAs displayed a distinct pattern of U > G > C > A (Appendix A).

### 3.3. Identification and Validation of Differentially Expressed miRNAs (DEMs)

Differential expression analysis identified 482 DEMs displaying sexually dimorphic expression patterns, comprising 179 testis-biased and 303 ovary-biased miRNAs (Figure 3). Among these, 13 DEMs exhibited exclusive ovarian expression while 17 were uniquely expressed in testicular tissue. To validate the expression profiles of these miRNAs, eight DEMs were randomly selected for RT-qPCR verification. The results showed that novel_miR_1049 (Ovarian-specific expression), novel_miR_1203, novel_miR_1006, and novel_miR_826 were significantly upregulated in the ovary (*p* < 0.05), whereas sar-miR-10d-1, novel_miR_579, novel_miR_110, and novel_miR_1112 were significantly upregulated in the testis (*p* < 0.05) (Figure 4) (Appendix A). The expression patterns detected by RT-qPCR were consistent with the sequencing results, confirming the accuracy and reliability of the sequencing data.

### 3.4. Joint miRNA and Transcriptome Analysis

Transcriptome profiling of the corresponding samples identified 9048 DEGs exhibiting sex-biased expression patterns in our previous studies [34]. Integrative analysis of DEMs and DEGs identified 462 DEMs potentially targeting 3340 DEGs through 3′UTR binding sites, forming 13,773 DEM–DEG regulatory pairs. Of these, 9403 pairs (68.27%) showed inverse differential expression trends, while 4370 pairs (31.73%) displayed consistent trends (Appendix A). Functional enrichment analysis of the 3340 DEGs identified significant associations with two GO terms and 18 KEGG pathways (Appendix A). These DEGs were particularly enriched in several pathways related to gonadal development and function, including the cAMP signaling pathway, Rap1 signaling pathway, MAPK signaling pathway, oxytocin signaling pathway, oocyte meiosis, GnRH secretion, and cGMP-PKG signaling pathway (Figure 5).

### 3.5. Regulatory Relationships Between miRNAs (DEMs) and Sex-Biased Genes (DEGs)

In the ovary, novel_miR_1351 exhibited the highest expression level among the DEMs, and was predicted to target 31 DEGs. Among these 31 DEGs, the majority (25 genes, 80.65%) displayed testis-biased expression, including *Rock1*, *Hk2*, *Thrb*, and *Par3*, while a smaller proportion (6 genes, 19.35%) showed ovarian-biased expression, such as *Frk*, *Mrpl21*, and *Elovl7*. Notably, certain target genes may play a role in the process of gonadal development. For instance, *Rock1* is a component of the TGF-β, cAMP, cGMP-PKG, and oxytocin signaling pathways [6,44,45,46], which are closely associated with gonadal development and sex determination. Additionally, *Taok1*, *Taok2*, and *Frk* are members of the MAPK signaling pathway, which has also been shown to be involved in the regulation of gonadal development [47] (Figure 6). In contrast, sar-miR-143-3p exhibited a distinct expression pattern, showing the highest expression level in the testis. This miRNA was predicted to target 27 DEGs, of which the majority (24 genes, 88.89%) displayed testis-biased expression, including *Camk4*, *Kcmal*, *Rab8a*, and *Slc6a4*, while only 3 genes (*Rgs3*, *Ndufaf6*, and *Tnfsf14)* showed ovarian-biased expression (Figure 7).

Based on previous studies of *S. argus*, 21 key sex-related genes were screened for their potential regulatory relationships with miRNAs (Appendix A). The results showed that three male development-associated genes (*Gsdf*, *Hsd3b7*, and *Bmp8*) (DEGs) demonstrated binding sites with 6 DEMs, while six female development-related genes (*Zar1*, *Zar1l*, *Gdf9*, *Hsd17b12*, *Sox3*, and *Foxl2*) (DEGs) exhibited binding sites with 42 DEMs. Among these interactions, 9 DEM–DEG pairs showed inverse differential expression trends, whereas 39 DEM–DEG pairs displayed consistent trends (Figure 8). Among them, miRNAs showed an inverse expression trend with *Foxl2*/novel_miR_110, *Gdf9*/novel_miR_802 and *Gdf9*/novel_miR_1263, while miRNAs showed the same expression trend with *Gsdf*/sar-miR-143-5p-4, *Gsdf*/sar-miR-143-5p-5 and *Sox3*/novel_miR_379 showed the same expression trend.

## 4. Discussion

At present, high-throughput sRNA sequencing has emerged as a prevalent methodology for miRNA identification. This advanced approach has enhanced the discovery and characterization of a substantial number of miRNAs across diverse tissues in various fish species. In this study, we delineated the regulatory networks between miRNAs and their target genes in *S. argus* gonads through an integrated analysis of DEMs and DEGs utilizing next-generation sequencing technology. The findings yield novel insights into the potential regulatory mechanisms of miRNAs in regulating gene expression during gonadal development, including sex determination and differentiation in *S. argus*. Moreover, these data establish a molecular foundation for advancing sex-controlled breeding strategies in *S. argus*.

Our analysis identified a total of 2210 miRNAs in *S. argus*, of which 1431 were novel miRNAs, representing a substantial proportion (64.75%) of the total miRNAs. Comparative analysis revealed interspecies variation in miRNA profiles across different fish species. For example, *Trachinotus ovatus* (*T. ovatus*) exhibited 279 miRNAs (100 novel, 35.84%), *Acipenser schrencki* (*A. schrenckii*) harbored 730 miRNAs (51 novel, 6.99%), and *Trachinotus blochii* (*T. blochii*) contained 1453 miRNAs (69 novel, 4.81%), demonstrating divergence in miRNA composition [48,49,50]. Notably, *S. argus* demonstrated both the highest number of DEMs and the greatest proportion of novel miRNAs among the four analyzed species. It is particularly relevant to mention that the identification of DEMs in *S. argus* was conducted using more stringent screening criteria (Appendix A). However, comparisons of miRNA sequencing data across these four species may be influenced by differences in experimental conditions, sequencing depth, and annotation methods. Notably, despite these considerations, sar-miR-214-3p-2 emerged as a conserved miRNA across all species, albeit with distinct expression patterns: exhibiting testis-biased expression in *S. argus* (identified as a DEM in our study), ovary-biased expression in *T. ovatus*, and no significant sex-biased expression in *A. schrenckii* or *T. blochii*. In *S. argus* specifically, we identified 117 potential target genes of sar-miR-214-3p-2, including 6 genes associated with sex-related signaling pathways (Appendix A). These findings, while highlighting potential species-specific regulatory roles, underscore the need for standardized comparative approaches and functional validation to determine whether miR-214-3p-2′s regulatory function is evolutionarily conserved among teleosts.

Emerging evidence reveals that miRNA-mediated negative regulation plays a fundamental role in sexual development across diverse species. In the gonad of the *S. argus*, a high proportion of predicted DEMs demonstrate inverse expression patterns relative to their target DEGs. This observation is consistent with established DEM–DEG regulatory mechanisms reported in previous studies [51,52]. This inverse correlation suggests that highly expressed miRNAs may function through translational inhibition or mRNA degradation of their target mRNAs, thereby downregulating gene expression, whereas lowly expressed miRNAs exhibit weakened translational repression or degradative effects on their target genes. Typically, miRNAs exert their regulatory effects by binding to target mRNAs, establishing a negative regulatory relationship. For instance, in *Paralichthys olivaceus* (*P. olivaceus*), miR-202-5p exhibits marked testis-specific expression, showing significantly higher abundance in male gonads compared to ovarian tissues. Molecular characterization revealed that miR-202-5p directly targets *Cbx2*, a key transcriptional regulator of gonadal development. This miRNA–mRNA interaction establishes a clear negative regulatory relationship, wherein miR-202-5p-mediated suppression of *Cbx2* appears to facilitate spermatogenic processes [53]. A conserved regulatory paradigm was identified in the *B. dorsalis*, where miR-1-3p demonstrates sexually dimorphic expression, being preferentially expressed in male embryos during the critical sex determination period. Comprehensive functional analyses, including target prediction and experimental validation, confirmed that miR-1-3p negatively regulates the master sex-determining gene *Bdtra*. Through a combination of embryonic microinjection and gene editing approaches, researchers conclusively demonstrated that miR-1-3p serves as a critical upstream modulator of male sex determination in this dipteran species [20].

In the ovaries of the *S. argus*, novel_miR_1351, a member of the miR-10 family, demonstrates the highest expression level among all differentially expressed miRNAs (DEMs). Furthermore, the genes predicted to be targeted by novel_miR_1351 are significantly enriched in pathways associated with gonad development (Figure 6), suggesting a potential regulatory role of this miRNA in ovarian function and development. This finding aligns with previous studies demonstrating that the miR-10 family plays critical regulatory roles in ovarian granulosa cells (GCs) across various species, including human, mouse, rat, and pig. miR-10a and miR-10b have been demonstrated to regulate the proliferation and apoptosis of GCs by targeting *Bdnf* [54]. Similarly, in pig ovarian GCs, miR-10a-5p has been shown to suppress steroid hormone synthesis through its targeting of *Creb1* [55]. These findings highlight the conserved and multifaceted regulatory roles of the miR-10 family in ovarian function across species. In the present study, although *Bdnf* and *Creb1* were not identified as DEGs, target prediction analysis using miRanda revealed that *Creb1* is potentially regulated by 10 miRNAs. These include novel_miR_380 and novel_miR_1227 (belonging to the miR-263 family), as well as novel_miR_125, novel_miR_1342, novel_miR_1396, novel_miR_175, novel_miR_246, novel_miR_407, novel_miR_511, and novel_miR_619. Notably, among these, only novel_miR_246 was identified as a DEM, exhibiting higher expression in the testis compared to the ovary. This suggests a potential tissue-specific regulatory role of novel_miR_246 in modulating *Creb1* expression. However, no miRNA–mRNA targeting relationships were predicted for *Bdnf* in the *S. argus*, suggesting that its regulation in this species may involve alternative mechanisms or remain unidentified under the current analysis framework. On the other hand, sar-miR-143-3p is the most highly expressed miRNA in the testis of the *S. argus*. Previous studies have identified *Kras* and *Fshr* as target genes of miR-143 in mouse and bovine GCs, respectively. Overexpression of miR-143 has been shown to downregulate the expression of *P450scc*, *3β-HSD*, and *StAR*, leading to reduced levels of progesterone (P4) and estradiol (E2) in bovine GCs, whereas inhibition of miR-143 produces the opposite effect [56,57]. Notably, the sequence of sar-miR-143-3p is identical to that of mouse miR-143 (Mmu-miR-143-3p), while bovine miR-143 (bta-miR-143) differs by an additional G at the 3′ end. However, in this study, neither *Kras* nor *Fshr* were predicted as target genes of sar-miR-143-3p (Figure 7). Instead, *Kras* was predicted to be regulated by novel_miR_1430, novel_miR_288, novel_miR_292, and novel_miR_1215, while *Fshr* was predicted to be targeted by novel_miR_108, novel_miR_1216, novel_miR_238, novel_miR_482, novel_miR_58, novel_miR_635, and novel_miR_649. These findings suggest potential species-specific differences in miRNA-mediated regulatory networks.

The DEMs with target sites on critical genes involved in sex determination and differentiation were also predicted in the *S. argus*. As mentioned earlier, *Foxl2* is a key gene in the female pathway. In mice, *Foxl2* expression in ovarian granulosa cells is negatively regulated by miR-133b, influencing granulosa cell function and follicular development [58]. In *O*. *niloticus* gonads at 5 days after hatching (dah), a DEM (miR-7977) has been predicted to regulate *Foxl2* [59]. In the *S. argus*, the predicted regulatory DEM for *Foxl2* is novel_miR_110, which differs from the miRNAs identified in mice and *O*. *niloticus*. Another crucial gene in the female pathway is *Cyp19a1*/*Cyp19a1a*. In pigs, miR-10b has been shown to regulate *Cyp19a1*, inhibiting its expression and function in granulosa cells [60]. In *O*. *niloticus* gonads at 5 dah, a DEM, miR-30a, has been predicted to regulate *Cyp19a1a* [59]. Similarly, in the *E. coioides*, miR-26a-5p, which is highly expressed in the testis, can downregulate *Cyp19a1a* expression [21]. However, no regulatory miRNAs for *Cyp19a1* were predicted in the *S. argus*. These observations indicate that the regulatory miRNAs for *Cyp19a1* vary significantly among pigs, *O*. *niloticus*, and *E. coioides*. Collectively, these findings underscore the species-specific nature of miRNA regulatory mechanisms governing female pathway-related genes, highlighting the diversity and complexity of miRNA-mediated regulation across different species.

One key objective of this study was to identify miRNAs that might regulate *Dmrt1Y*, a sex-determining gene in *S. argus* [31,32]. While our bioinformatic analyses failed to predict any miRNAs targeting *Dmrt1Y* in this species, computational studies have identified potential *Dmrt1*-regulating miRNAs in several other fish species. These include miR-212 (a DEM) in *O. niloticus* and multiple DEMs (asc-miR-159a, 2779, 2779-1, and 203b-3p) in *A. schrenckii* [48,59]. Notably, in *Takifugu rubripes* (*T. rubripes*), experimental validation has confirmed that fru-miRNA-122 (a DEM) directly downregulates *Dmrt1* expression [14]. *Gsdf*, a gene downstream of *Dmrt1* in the *S. argus* [61], is predicted to be targeted by sar-miR-143-5p-4 and sar-miR-143-5p-5 in this species. Similarly, the *T. rubripes* suggest that Gsdf may be regulated by miR-730, fru-miR-216b, and several DEMs, including novel-m0524-5p, m0272-5p, m0408-5p, and m0087-3p [14]. Another gene involved in gonad development, *Amh*, exhibits male-biased sexual dimorphism in expression. In *Cyprinus carpio* (*C*. *carpio*); miR-153b-3p, which shows sexually dimorphic expression in the gonads; inhibits *Amh* expression, thereby modulating male germ cell proliferation and differentiation during spermatogenesis [13]. Bioinformatic predictions identified DEMs (miR-456 and miR-138) as potential regulators targeting *Amh* in 60 dah *O. niloticus* [62], while DEMs (miR-96 and miR-449) were predicted to be regulatory factors of *Amh* in 5 dah *O. niloticus* [59]. However, in the *S. argus*, no miRNAs were predicted to regulate *Amh*. These results highlight the lack of conservation in miRNA-mediated regulation of genes involved in sex determination and differentiation across fish species, underscoring the diversity and complexity of these regulatory networks.

The observed interspecies differences in miRNA regulation of the male sex-determination gene *Dmrt1Y* and other gonad development-related genes (*Foxl2*, *Cyp19a1*/*Cyp19a1a*, *Gsdf*, and *Amh*) in *S. argus* may arise from both technical and biological factors. Since the predicted regulatory relationships between these genes and DEMs have not been experimentally validated in either this study or some comparative studies, the observed variations could result from technical differences in miRNA and 3′UTR sequencing analyses (including sequencing platforms, library preparation methods, and bioinformatics pipelines), as well as discrepancies in target prediction algorithms and their parameter settings. When the predicted gene–DEM regulatory pairs exhibit minimal technical artifacts or have been experimentally validated across species, these observed differences may reflect species-specific characteristics in the miRNA regulatory mechanisms of gonad development-related genes from a biological perspective. In cases where highly conserved miRNAs are identified among the compared species, the differential gene–DEM regulatory relationships may result from species-specific variations in the 3′UTRs of conserved genes. Conversely, when miRNAs show low sequence similarity across species, the observed interspecies differences could originate from either the miRNAs themselves, the 3′UTRs of conserved genes, or both exhibiting species-specific features.

## Figures and Tables

**Figure 1 animals-15-01564-f001:**
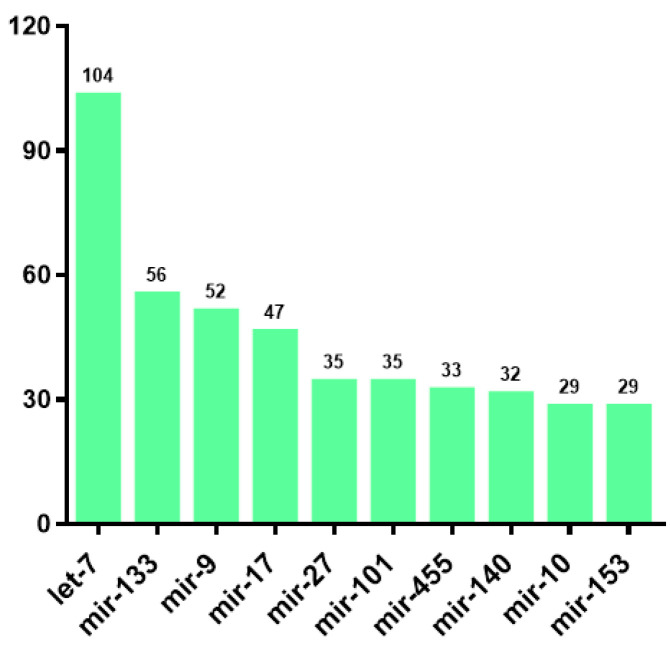
Statistics of the top 10 most abundant miRNA families.

**Figure 2 animals-15-01564-f002:**
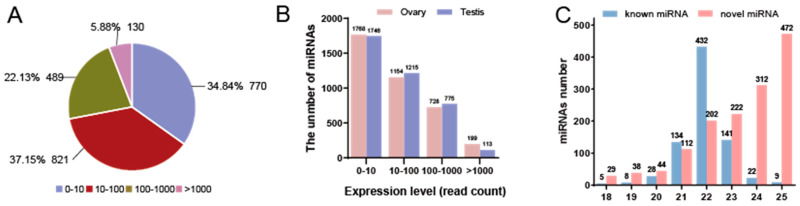
Overview of miRNA expression characteristics in gonads of *S. argus* using high-throughput sequencing. (**A**) Proportion and number of miRNAs in different expression levels classified by read count. (**B**) Comparison of the number of miRNAs in different groups classified by read number in ovary and testis. (**C**) Length distribution of known miRNA and novel miRNA.

**Figure 3 animals-15-01564-f003:**
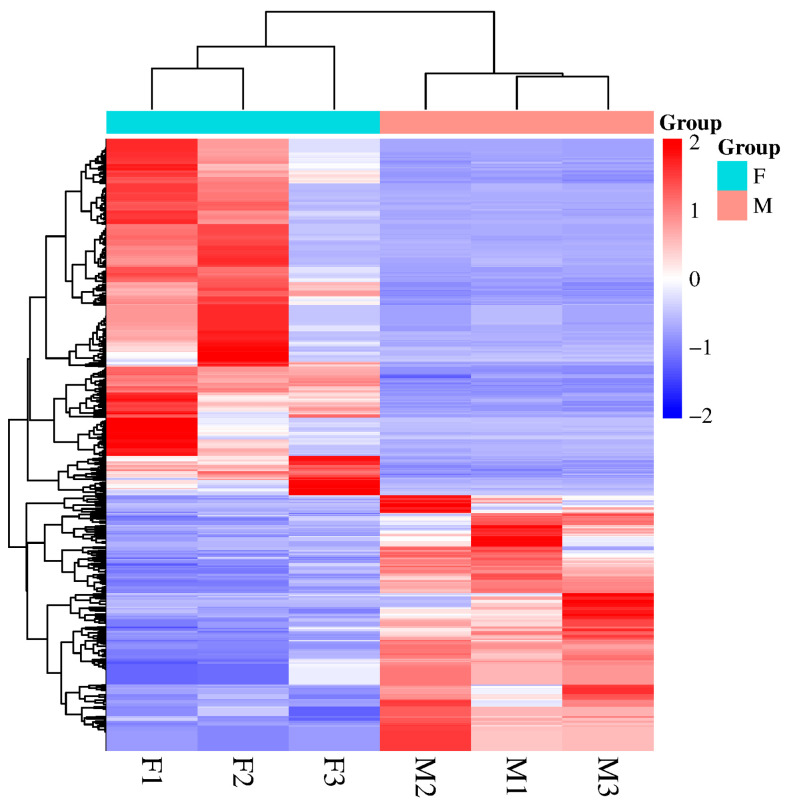
Heat map of DEMs between testis and ovaries of spotted scat. A total of 482 DEMs were identified with |log2(FC)| ≥ 2 and a corrected *p* < 0.01. Rows represent different miRNAs, while columns correspond to testis (M) and ovary (F). The expression data for each miRNA were derived from three biological replicates.

**Figure 4 animals-15-01564-f004:**
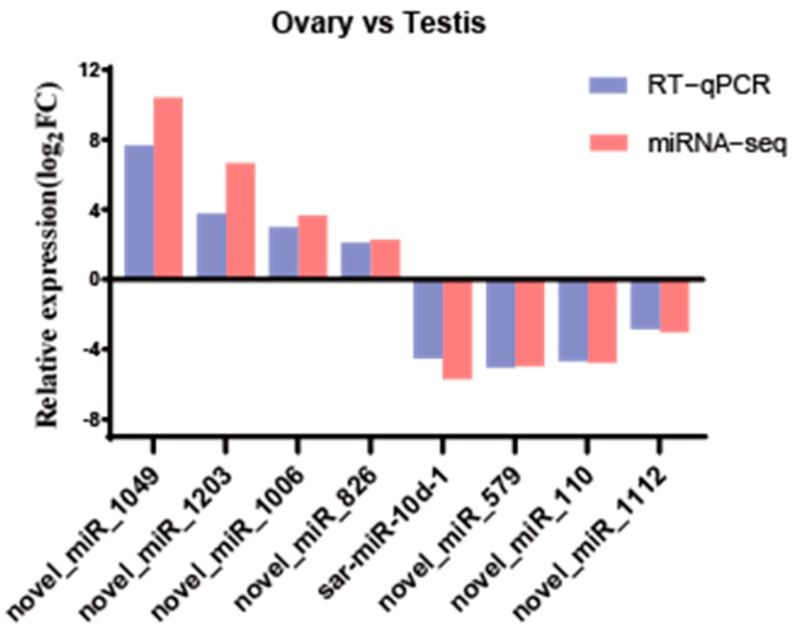
Comparison of expression levels of eight DEMs using miRNA-seq and qRT-PCR. To calculate the log2FC (log2 fold change) between male and female gonadal samples using the 2^−ΔΔCt^ method from RT-qPCR data (*n* = 3).

**Figure 5 animals-15-01564-f005:**
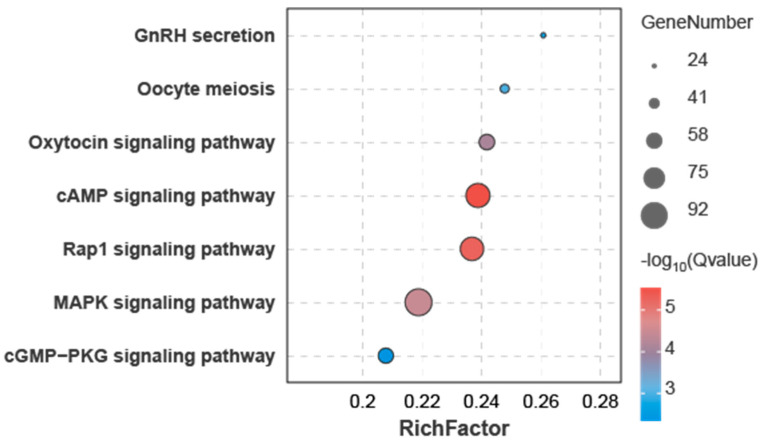
Gonadal DEMs target KEGG pathways enriched by DEGs.

**Figure 6 animals-15-01564-f006:**
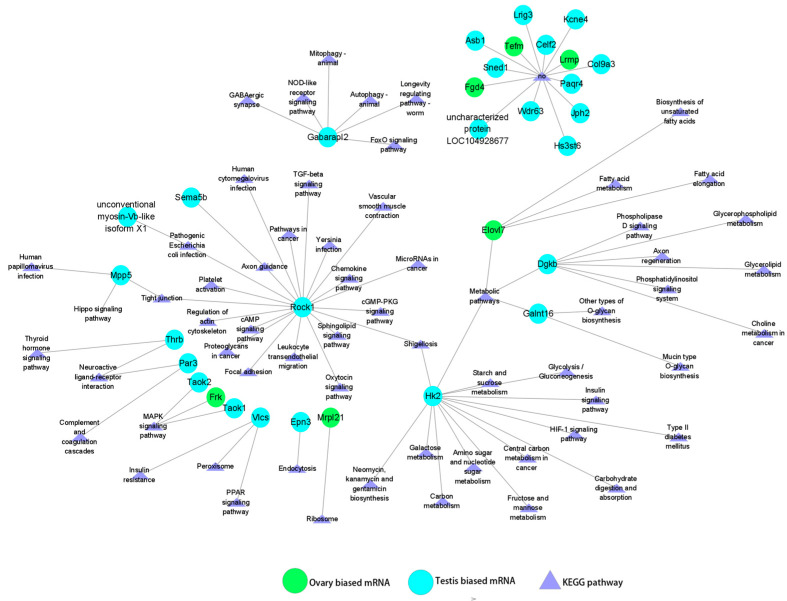
Target prediction of DEGs and KEGG analysis of novel_miR-1351, the most abundant DEM in ovary.

**Figure 7 animals-15-01564-f007:**
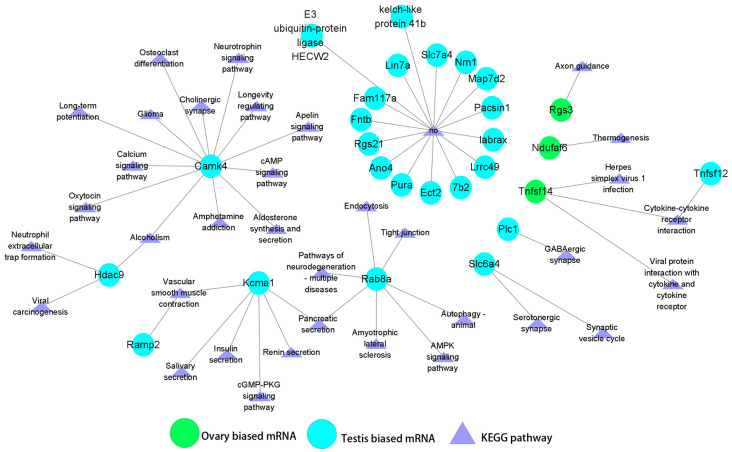
Target prediction DEGs and KEGG analysis of sar-miR-143-3p, the most abundant DEM in testis.

**Figure 8 animals-15-01564-f008:**
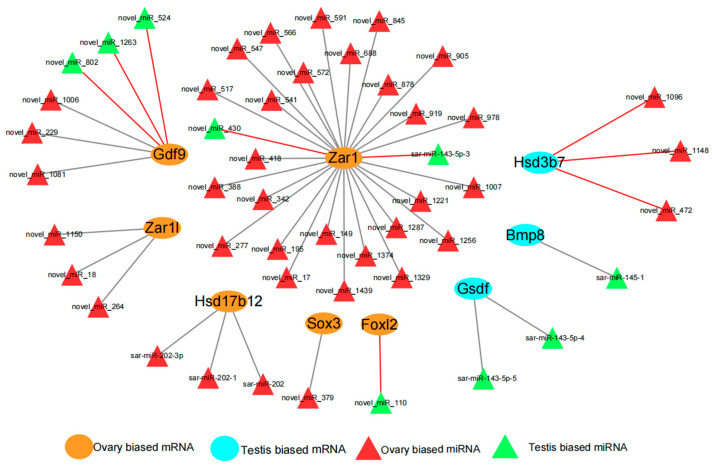
The interacted regulatory network between sex-related target genes and differentially expressed miRNAs. Black lines indicate that miRNAs and mRNAs show the same bias in differential expression in the gonad, while red lines indicate the opposite bias.

**Table 1 animals-15-01564-t001:** Primer sequences of miRNAs for the qPCR analysis.

miRNA	Primers (5′–3′)
novel_miR_1049	GCTGTTCGGCTGAGCCTC
novel_miR_1203	AAAGAGCACAGTGGCCTCCATAATC
novel_miR_1006	TCTCTGATGCGGGCCATGACCAGTC
novel_miR_826	TGCTGTGGTAGCCATGCTGGTT
sar-miR-10d-1	TACCCTGTAGAACCGAATGTGT
novel_miR_579	TCGTGTCTTGTGTTGCAGCCAGT
novel_miR_110	TCCTCATTGTGCATGCTGTGTG
novel_miR_1112	TGTGATATTGTTTCATGTGATTCG
Universal primer R	GATCGCCCTTCTACGTCGTAT
U6F	GCCACTTCGGCAGCACATAC
U6R	TTGCGTGTCATCCTTGC

**Table 2 animals-15-01564-t002:** Classification of the obtained small RNAs.

Indexes	F1(%)	F2(%)	F3(%)	M1(%)	M2(%)	M3(%)
Raw reads	18,797,843	11,972,637	11,847,428.5	9,939,298	11,847,428.5	12,599,932
Clean reads	18,690,270 (100%)	11,935,960 (100%)	11,774,117 (100%)	9,888,937 (100%)	11,795,830 (100%)	12,518,068 (100%)
rRNA	1,261,728 (6.75%)	435,138 (3.65%)	395,408 (3.36%)	231,912 (2.35%)	155,958 (1.32%)	192,147 (1.53%)
snoRNA	13,422 (0.07%)	8637 (0.07%)	1671 (0.01%)	1100 (0.01%)	1087 (0.01%)	1456 (0.01%)
tRNA	103,138 (0.55%)	71,488 (0.60%)	38,397 (0.33%)	92,592 (0.94%)	179,553 (1.52%)	139,963 (1.12%)
Repbase	248,340 (1.33%)	164,456 (1.38%)	123,269 (1.05%)	27,764 (0.28%)	30,414 (0.26%)	36,912 (0.29%)
Unannotated	17,063,642 (91.30%)	11,256,241 (94.30%)	11,215,372 (95.25%)	9,535,569 (96.42%)	11,428,818 (96.89%)	12,147,590 (97.05%)
Mapped reads	12,561,162	8,134,819	8,376,643	6,582,447	7,773,622	8,573,809

Raw reads (Sequencing raw data), Clean reads (High-quality sequences), Ribosomal RNAs (rRNAs), Transporter RNAs (tRNAs), Small nucleolar RNAs (snoRNAs), Repbase, Unannotated reads (containing miRNAs), Mapped reads (Unannotated reads compared to the reference genome).

## Data Availability

The data that support the findings of this study are available upon reasonable request. The raw reads used in this article have been deposited into the Sequence Read Archive (SRA) of the NCBI database under BioProject accession number: PRJNA1149578.

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
