# Peer review of "Integrated Analysis of Differential Expression Profiles of miRNA and mRNA in Gonads of Scatophagus argus Provides New Insights into Sexually Biased Gene Expression"

_animals, 2025, doi:10.3390/ani15111564_

Round 1
Reviewer 1 Report
Comments and Suggestions for Authors
Ya-Ling Lei, Kai-Zhi Jiao, Yuan-Qing Huang, Yu Li, Yu-Wei Wu, Gang Shi, Hong-Juan Shi, Hua-Pu Chen, Si-Ping Deng, Guang-Li Li, Dong-Neng Jiang, Wen-Jing Tao
Integrated analysis of differential expression profiles of miRNA and mRNA in gonads of Scatophagus argus provides new insights into the sexually biased gene expression
animals-3551547
This work analyzes microRNAs and mRNA sequences in the gonads of Scatophagus argus, an economically relevant species in southern China, obtained using high-throughput sequencing. Several potential miRNAs were examined using RT-qPCR, and the results were compared with the expression levels detected using miRNA-Seq data. The expression levels of miRNAs and mRNA sequences were compared to identify potential regulatory relationships. In addition, a comparative analysis compared miRNA profiles among different fish species. This study provides interesting insights into the regulatory roles of miRNA sequences and sex-related genes.
The methods presented are adequate for studying sex differentiation in this species.
My recommendation is that the manuscript should be reconsidered after major revision.
MAJOR COMMENTS
- In my opinion, the weakest point of this manuscript is the comparative analysis of miRNA profiles among different fish species. Although it is obviously interesting and necessary to compare the results with those of other species, potential differences among the experimental conditions make it difficult to reach strong conclusions. I do not think that statements as "Notably, S. argus demonstrated both the highest number of DEMs and the greatest proportion of novel miRNAs among the four analyzed species" and "These findings suggest that the miRNA-mediated regulatory network in S. argus gonads may possess a greater degree of complexity than previously recognized" are particularly supported by the data.
- L142-145 "The samples used in this experiment are the same as those were used for the integrated analysis of the gonadal methylome and transcriptome by Jiao et al (2024). Both male and female gonads are in stage IV according to the histological observation [26]" To be able to judge how similar or different the samples were, it is important to provide additional information about the samples (i.e. sampling dates, weigh and length). Histological sections are shown in Jiao et al (2024), but I did not find information about the size of the samples.
- Section "2.6. Verification of miRNA Expression by Real-Time Quantitative PCR (RT-qPCR)" In my opinion, additional details are required about how primers were designed to verify miRNA expression and the sequence that is being targeted (i.e. the mature miRNA or the precursor sequence).
- L267-270 "Among these miRNAs, 1,161 miRNAs were annotated and classified into 218 families, and the ten most abundant miRNA families were let-7, mir-133, mir-9, mir-17, mir-27, mir-455, mir-140, mir-10, mir-30, and mir- 153 (Figure 1)"
How were the 1,161 miRNAs classified into families? Please describe the procedure.
MINOR COMMENTS
- L80-83 (and other instances through the manuscript), the scientific name of species is usually written in its full form at the first mention and abbreviated thereafter. Here, you first write the abbreviated name and between parentheses the full name. I wonder if you have a particular reason to do this. Otherwise, I would recommend that you follow the convention I mentioned before.
- L141-142 "Sex and gonadal stage were determined by H&E staining using established histological methods [25]" I would write "H&E" in full (i.e. hematoxylin and eosin) to help readers not familiar with these methods. The reference provided does not appear to be a general reference for these histological methods.
- L180-186, the Bowtie version mentioned in the manuscript is quite old. Please double-check that this was the version used.
- L223-224 "The relative expression levels of miRNAs were quantified using the 2 -ΔΔCt method, with U6 serving as the internal reference gene"
Are there previous studies supporting the use of this gene in this species as internal reference? Did you consider using multiple reference genes?
- L273-275 "The length distribution analysis revealed that miRNAs were predominantly between 10-100 bp (37.15%, n=821), followed by 0-10 bp (34.84%, n=770), while miRNAs exceeding 1000 bp (5.88%, n=130) represented the least abundant population (Figure 2A)" This sentence seems to be incorrect. Figure 2A presents the expression level (read number) categories vs. number of miRNAs, not the miRNA length distribution.
- L277-278 "In detail, the majority of length for known miRNAs was 22 nucleotides (nt/bp), whereas newly predicted miRNAs showed a peak at 25 nt (Figure 2C)"
I suggest to just use "nt" between the parenthesis.
- L293-296 "Furthermore, differential expression analysis of these miRNAs identified 482 DEMs, including 179 with testicular bias and 303 with ovarian bias (Figure 4). Among these DEMs, 13 were ovary-specific, while 17 were expressed in both testis and ovary"
These numbers are not clear to me. If there were 482 DEMs and 13 were ovary-specific, shouldn't the remaining miRNAs exhibit expression in both testis and ovaries (i.e. 482 - 13 = 469)? Are there any miRNAs with testis-specific expression?
- L455-456 "In O. niloticus gonads at 5 days after hatching (dah), a DEM (miR-7977) has been predicted to regulate Foxl2 [50](Tao et al., 2016)"
Correct "... Foxl2 [50](Tao et al., 2016)" to "Foxl2 [50]"
- Reference 19. Zhang, X.; Li, L.; Jiang, H.; Ma, J.E.; Li, J.; Chen, J. Identification and differential expression of microRNAs in testis and ovary of Amur sturgeon (Acipenser schrenckii). Gene 2018, 658, 36–46. doi:10.1016/j.gene.2018.03.014.
"Acipenser schrenckii" should be italicized.
- Figure 3. This figure does not add any extra information to that presented in the text. My recommendation is to remove this figure.
- Figure 4. This figure is missing.
- Figure 5. Please state the specific type of error bar used in the caption.
- Figure 6. The caption for Figure 6 is on page 10, but the figure is on page 9.
- Figure S1 (caption). Change "S. argu" to "S. argus"
- Table 1. The table is broken between two pages.
- Supplementary Table 4 (S4.xlsx): Typo. In the headers change "Geng name" to "Gene name".
Author Response
Response to Reviewer 1 Comments
MAJOR COMMENTS
In my opinion, the weakest point of this manuscript is the comparative analysis of miRNA profiles among different fish species. Although it is obviously interesting and necessary to compare the results with those of other species, potential differences among the experimental conditions make it difficult to reach strong conclusions. I do not think that statements as "Notably, S. argus demonstrated both the highest number of DEMs and the greatest proportion of novel miRNAs among the four analyzed species" and "These findings suggest that the miRNA-mediated regulatory network in S. argus gonads may possess a greater degree of complexity than previously recognized" are particularly supported by the data.
Reply:
We sincerely appreciate the reviewer's constructive criticism regarding the cross-species comparison. To address these concerns, we have made the following changes and clarifications:
Deleted "These results suggest that the miRNA-mediated regulatory network in the gonads of S. argus may possess a greater degree of complexity than previously recognised", this conclusion and explains that these species comparisons are subject to an error factor L430-432 "However, comparisons of miRNA sequencing data across these four species may be influenced by differences in experimental conditions, sequencing depth and annotation methods".
L142-145
"The samples used in this experiment are the same as those were used for the integrated analysis of the gonadal methylome and transcriptome by Jiao et al (2024). Both male and female gonads are in stage IV according to the histological observation [26]" To be able to judge how similar or different the samples were, it is important to provide additional information about the samples (i.e. sampling dates, weigh and length). Histological sections are shown in Jiao et al (2024), but I did not find information about the size of the samples.
Reply:
We have added detailed sample information (weight, length, and sampling date) in Supplementary Table 1 .
Table 1. Samples Information
Code |
F1 |
F2 |
F3 |
M1 |
M2 |
M3 |
Weight(g) |
358 |
360 |
370 |
226 |
228 |
243 |
Length(cm) |
21.5 |
22.0 |
22.4 |
19.7 |
19.4 |
20.0 |
Sampled July 14, 2019; all gonads at stage IV
Section "2.6. Verification of miRNA Expression by Real-Time Quantitative PCR (RT-qPCR)" In my opinion, additional details are required about how primers were designed to verify miRNA expression and the sequence that is being targeted (i.e. the mature miRNA or the precursor sequence).
Reply:
We sincerely appreciate the reviewer's constructive suggestions. In response to these comments, we have revised the RT-qPCR methodology section (L 252-256) to provide more precise technical details: "Quantification was performed using the poly(A)-tailing-based RT-qPCR method. This approach employed miRNA-specific forward primers designed against mature sequences and a universal reverse primer (R: GATCGCCCTTCTACGTCGTAT) complementary to the 3'-adapter sequence [42]. The forward primers were designed using the online primer design system from Sangon Biotech (Shanghai, China) (https://store.sangon.com/newPrimerDesign)"
L267-270
"Among these miRNAs, 1,161 miRNAs were annotated and classified into 218 families, and the ten most abundant miRNA families were let-7, mir-133, mir-9, mir-17, mir-27, mir-455, mir-140, mir-10, mir-30, and mir- 153 (Figure 1)"
How were the 1,161 miRNAs classified into families? Please describe the procedure.
Reply:
We have clarified the miRNA family classification method in the revised manuscript, L238-240 "This analysis required perfect matching of the seed sequence (2-8 nt) to classify miRNAs as belonging to the same family, leveraging sequence similarity for accurate annotation and classification."
L80-83
(and other instances through the manuscript), the scientific name of species is usually written in its full form at the first mention and abbreviated thereafter. Here, you first write the abbreviated name and between parentheses the full name. I wonder if you have a particular reason to do this. Otherwise, I would recommend that you follow the convention I mentioned before.
Reply:
Thank you for your positive comments on our draft. Your suggestions were adopted. As suggested, we have standardized the formatting of species names throughout the manuscript, ensuring all first mentions follow the required nomenclature guide.
L141-142
"Sex and gonadal stage were determined by H&E staining using established histological methods [25]" I would write "H&E" in full (i.e. hematoxylin and eosin) to help readers not familiar with these methods. The reference provided does not appear to be a general reference for these histological methods.
Reply:
We sincerely appreciate the reviewer’s constructive feedback. In response to the comments, we have (1) expanded "H&E" to "hematoxylin and eosin" in L157 and revised and updated reference as suggested.
[34] Fischer, A.H.; Jacobson, K.A.; Rose, J.; Zeller, R. Hematoxylin and eosin staining of tissue and cell sections. CSH Protoc 2008, pdb.prot4986. doi:10.1101/pdb.prot4986.
L180-186
-the Bowtie version mentioned in the manuscript is quite old. Please double-check that this was the version used.
Reply:
We confirm Bowtie v1.0.0 was used for consistency with our validated pipeline and to ensure reproducibility. Though older, it remains effective for our miRNA alignment tasks
.L223-224
"The relative expression levels of miRNAs were quantified using the 2 -ΔΔCt method, with U6 serving as the internal reference gene"
Are there previous studies supporting the use of this gene in this species as internal reference? Did you consider using multiple reference genes?
Reply:
We sincerely appreciate the reviewer's insightful comment. U6 was selected as the internal reference gene based on two key considerations: (1) its established use in prior miRNA studies of Scatophagus argus (Li et al., 2022; Leng et al., 2025), and (2) our experimental validation demonstrating excellent amplification efficiency (97.72%) and linearity (R²=0.9957). While we acknowledge that using multiple reference genes can provide more robust normalization, U6 has been widely adopted as a reliable single reference for miRNA quantification in this species.
Li Z, Guo Y, Ndandala CB, Chen H, Huang C, Zhao G, Huang H, Li G, Chen H. Analysis of circRNA and miRNA expression profiles in IGF3-induced ovarian maturation in spotted scat (Scatophagus argus). Front Endocrinol (Lausanne). 2022 Nov 25;13:998207. doi: 10.3389/fendo.2022.998207. PMID: 36506051; PMCID: PMC9732426.
Leng Z, Lin X, Zhai Y, et al. MiR-9-z/zp2, miR-3154-y/zp2, and miR-3533-y/zp3 participated in ovarian development and adaptation to salinity stress in spotted scat (Scatophagus argus)[J]. Aquaculture Reports, 2025, 40: 102653.
L273-275
"The length distribution analysis revealed that miRNAs were predominantly between 10-100 (37.15%, n=821), followed by 0-10(34.84%, n=770), while miRNAs exceeding 1000 bp (5.88%, n=130) represented the least abundant population (Figure 2A)" This sentence seems to be incorrect. Figure 2A presents the expression level (read number) categories vs. number of miRNAs, not the miRNA length distribution.
read number normalized analysis showed that miRNAs were mainly in the range of 10-100 (37.15%, n=821), followed by 0-10(34.84%, n=770), while miRNAs exceeding 1000 bp (5.88%, n=130) represented the least abundant population (Figure 2A)
Normalized read count analysis revealed that the majority of miRNAs (37.15%, n=821) exhibited expression levels in the 10-100 range, followed by the 0-10 range (34.84%, n=770). In contrast, highly abundant miRNAs (>1000 reads) constituted the smallest proportion (5.88%, n=130) (Figure 2A).
Reply:
We sincerely appreciate the reviewer's careful reading. You are absolutely correct - the original text mistakenly described miRNA length ranges when it should have referred to normalized expression values. We have carefully revised L319-322 to accurately reflect that these values represent the normalized expression ranges of miRNAs after quantification analysis.
L277-278
"In detail, the majority of length for known miRNAs was 22 nucleotides (nt/bp), whereas newly predicted miRNAs showed a peak at 25 nt (Figure 2C)"
I suggest to just use "nt" between the parenthesis.
Reply:
We sincerely appreciate the reviewer's positive feedback and suggestion. The modification has been made as recommended(L324).
L293-296
"Furthermore, differential expression analysis of these miRNAs identified 482 DEMs, including 179 with testicular bias and 303 with ovarian bias (Figure 4). Among these DEMs, 13 were ovary-specific, while 17 were expressed in both testis-ovary"
These numbers are not clear to me. If there were 482 DEMs and 13 were ovary-specific, shouldn't the remaining miRNAs exhibit expression in both testis and ovaries (i.e. 482 - 13 = 469)? Are there any miRNAs with testis-specific expression?
Reply:
We sincerely appreciate the reviewer's careful reading. We have identified an important inaccuracy in our original description. The text has been revised at L339-340" Strikingly, while 13 DEMs were exclusively expressed in ovarian tissue, 17 were exclusively expressed in testes tissues."
L455-456
"In O. niloticus gonads at 5 days after hatching (dah), a DEM (miR-7977) has been predicted to regulate Foxl2 [50](Tao et al., 2016)"
Correct "... Foxl2 [50](Tao et al., 2016)" to "Foxl2 [50]"
Reply:
We have deleted the (Tao et al., 2016) (L507).
Reference 19. Zhang, X.; Li, L.; Jiang, H.; Ma, J.E.; Li, J.; Chen, J. Identification and differential expression of microRNAs in testis and ovary of Amur sturgeon (Acipenser schrenckii). Gene 2018, 658, 36–46. doi:10.1016/j.gene.2018.03.014.
"Acipenser schrenckii" should be italicized.
Reply:
We have italicized Acipenser schrenckii as suggested (L697)
Figure 3. This figure does not add any extra information to that presented in the text. My recommendation is to remove this figure.
Reply:
We sincerely appreciate the reviewer's valuable suggestion regarding Figure 3. After careful consideration, we agree that the figure's content was not essential to our core findings and have therefore removed it from the manuscript. All subsequent figure numbers have been updated accordingly.
Figure 4. This figure is missing.
Reply:
We sincerely apologize for the oversight regarding Figure 4. We have now: Reinstated Figure 4 in the manuscript (currently appears as Figure 3 after removal of the previous Figure 3)
Figure 5. Please state the specific type of error bar used in the caption.
Reply:
We have revised the Figure 4 caption to enhance methodological clarity while maintaining conciseness(L354-356). The updated version now specifies the use of three biological replicates (n = 3) and clearly indicates that fold changes (log2FC) were calculated using the 2−ΔΔCt method for qRT-PCR data. While we initially included error bars to represent mean ± SEM, upon further consideration we have removed them from the figure to improve visual clarity, as the key comparative data between miRNA-seq and qRT-PCR results remain clearly discernible. These modifications provide essential experimental details while optimizing the figure's presentation quality.
Supplementary Figure 2 Sexually dimorphic expression of DEMs validated by RT-qPC Relative expression levels of eight candidate miRNAs in male (M, blue) and female (F, red) gonads (n = 3 biological replicates). Data are presented as mean ± SD (error bars represent standard deviation of technical triplicates). Statistical significance was determined by two-tailed unpaired Student's t-test with Welch's correction (*p < 0.05, **p < 0.01, ***p < 0.001).
Figure 6. The caption for Figure 6 is on page 10, but the figure is on page 9.
Reply:
We have corrected the figure placement L369-371
Figure S1 (caption). Change "S. argu" to "S. argus"
Reply:
Corrected the species name from "S. argu" to "S. argus". (In Figure S2, the order of the tables has been adjusted.)
Table 1. The table is broken between two pages.
Reply:
Reformatted Table 1 to appear on a single page without breaks
Supplementary Table 4 (S4.xlsx): Typo. In the headers change "Geng name" to "Gene name".
Reply:
Fixed the header typo ("Geng name" → "Gene name"). (In Figure S 5, the order of the tables has been adjusted.)
Reviewer 2 Report
Comments and Suggestions for Authors
Overall, the author present a compelling dataset and analysis investigating the correlations between mRNA and microRNA in the gonads of the fish Scatophagus argus. The depth of the work seems sufficient and exploring the role of microRNAs in non-model organisms is certainly novel and worthy of investigation. However, there are a few issues that should be addresses in this manuscript before I would feel comfortable recommending for publication.
1) The scope of the citations feels quite narrow, especially in the introduction. I would encourage the authors to explore the literature and its significance to their work more broadly.
2) Methods need either more detail or clarity. For instance, what part of the gonad was sampled? Why did you use samples with low RIN values (in one part you said only sample >8.0 and in another >2.0)? This is a big deal as it could indicate the many of the microRNAs claimed to be found are actually degraded components of other RNA molecules. Knowing how this was avoided is key. How many cycles of PCR did you do during microRNA library construction? This is very important, especially for later when assessing differential expression. There is also very little information presented on how results are filtered and what criteria are for chosing to move forward with identifying what is a microRNA. The qPCR validation study is also not well explained. Why did you choose the targets you did? Why did you not consider differences in primer efficiencies among targets during qPCR? Clarifying this section and why you made the decisions you did would really help the reader understand
3) I am not sure I understand the logic behind the presentation of some of the results as well as where some data came from. For instance, you present that a large amount of your data was for RNA sequence >20 nt and it is not clear how this data was either considered or filtered out for later sections of the manuscript. Did you delete it? Seems pretty small for consideration in microRNA research. What does its prevalence say about the overall quality of your dataset? Also, what is going on with the sequence >1000 bp? Wasn't this NovoSeq data? How were the sequences so long? The validation study also should have been more expansive than 6 targets all >4 log(FC). Data analysis include samples down to 2 log FC and error bars on RNA-seq data in Figure 5. The lack of clarity here undermines your other compelling results.
4) The Joint miRNA/Transcriptome Analysis section would benefit from a bit more detail and some clarity of language as well. First, what is the level of confidence we should be associating with the 462 DEMs associatedw ith 3,340 DEGs and the potential 13,673 DEM-DEG pairs? This process seems rife false positives and seems to involve a fair amount of conjecture as presented. Could you provide some more information here to convince the reader that these are legitimate matches? I also found the whole section on negative and positive correlations of DEM-DEG pairs to be confusing and difficult to follow. As presented, it feels like the authors are saying that both cases hold equal value and either is equally likely to be expected. I would general think negative correlated expression would be more likely than positive. Regardless, it leaves the Discussion feeling full of conjecture. I think the manuscript would be strengthened by setting clear expectations and interpretations of results earily in the manuscript and then adhering to those expectations for downstream data analysis. Otherwise, the discussion ends up feeling a bit meandering. I found the comparative points to be the most interesting and compelling and would have loved to have seen more about that, especially some kind of explanation for why DMRT1 would be microRNA controlled in one organism and now another?
More less significant points:
A) Some grammatical mistakes with things like italics, use full Genus and species first time then abbreviate afterwards, and other minor errors. Please proofread.
B) Third paragraph of the intro is pretty generic. Consider revising it significantly.
C) I'm not sure what "bottlenecks" you are referring to in the paragraph of the intro
D) Figure 7 legend needs grammer work
E) Figure 3 feels unnecessary - would have been more interesting to see something describing miRNA families visually.
Author Response
Response to Reviewer 2 Comments
- The scope of the citations feels quite narrow, especially in the introduction. I would encourage the authors to explore the literature and its significance to their work more broadly.
Reply:
We sincerely appreciate the reviewer's constructive suggestion. We have thoroughly revised the introduction to broaden the literature coverage.
L54-57: We introduced the three types of fish sex determination mechanisms—genetic sex determination (GSD), environmental sex determination (ESD), and a mixed mode involving the synergistic regulation of both genetic and environmental factors.
L57-59: We highlighted the critical role of sex-determining genes in GSD.
L59-62: We described the signaling pathways associated with sex-determining genes.
L62-69: Consistent with the previous manuscript, we explained that sex determination involves a dynamic balance between male- and female-promoting pathway genes.
L69-72: As before, we highlighted that their precise regulatory networks and epigenetic mechanisms remain unclear.
L72-74: We revised the section on miRNA-mediated epigenetic regulation to better connect with the subsequent results and discussion.
L90-103: We refined the description of miRNAs and their roles in fish sex determination to strengthen logical coherence with the context.
L106: We added references [25,26] on high-throughput sequencing analysis, which serve as the foundation for the subsequent miRNA-mRNA joint analysis.
L107-117: We updated the literature to better describe the basis of miRNA-mRNA joint analysis in sex determination studies of other fish species.
L125-127: We adjusted the wording to clearly state the research bottleneck—"However, conventional hormonal treatments using androgen 17α-methyltestosterone (MT) and aromatase inhibitor letrozole (Le) fail to induce complete sex reversal in XX individuals [32]."
Newly added references:
[1,2] Literature on fish sex determination mechanisms
[3-5] Literature on sex-determining genes
[6,7] Literature on signaling pathways associated with sex-determining genes
[27-29] Updated references on miRNA-mRNA joint analysis in fish gonads
This revision enhances the depth and clarity of our study while ensuring better logical flow and stronger support from the literature.
- Methods need either more detail or clarity. For instance, what part of the gonad was sampled? Why did you use samples with low RIN values (in one part you said only sample >8.0 and in another >2.0)? This is a big deal as it could indicate the many of the microRNAs claimed to be found are actually degraded components of other RNA molecules. Knowing how this was avoided is key. How many cycles of PCR did you do during microRNA library construction? This is very important, especially for later when assessing differential expression. There is also very little information presented on how results are filtered and what criteria are for chosing to move forward with identifying what is a microRNA. The qPCR validation study is also not well explained. Why did you choose the targets you did? Why did you not consider differences in primer efficiencies among targets during qPCR? Clarifying this section and why you made the decisions you did would really help the reader understand.
Reply:
Thank you for your suggestion. We have substantially revised the Methods section to provide greater clarity and precision.
L150-155
Added detailed protocols for differential gonad sample processing: For RNA samples: in females, the gonadal capsule was removed and the central parenchymal tissue was collected (50-100 mg/tube); in males, the entire gonadal tissue including the capsule was collected. All RNA samples were flash-frozen in liquid nitrogen and stored at -80°C. For histological samples: gonadal tissues with intact capsules were cut into standardized 5×5×3 mm cubic blocks, fixed in Bouin's solution for 24 hours, followed by paraffin embedding and sectioning.
L170-174
(Modify issue regarding RIN values, previously misdescribed.)Corrected total RNA quality standards for library construction: total quantity ≥1.5 μg (sufficient for two library constructions), concentration ≥200 ng/μL, volume ≥10 μL, with OD260/280 ratio of 1.7-2.5, OD260/230 ratio of 0.5-2.5, normal 260 nm absorption peak, RIN value ≥8, and 28S/18S ratio ≥1.0 (preferably ≥2.0) with flat or minimally elevated baseline.
L199-332
Refined miRNA identification workflow: We first filtered sRNA sequencing data against known sRNA databases to remove non-miRNA sequences, retaining only unannotated potential miRNA precursors. These sequences were mapped to the reference genome and compared against known miRNA databases to identify annotated miRNAs. For remaining unmapped reads, we extended their genomic positions (±200bp upstream/50bp downstream) to capture potential precursor structures, then analyzed them using miRDeep2 (v2.0.5): (1) Mapper module aligned reads to define pre-miRNA boundaries (position/abundance/mismatches), (2) Quantifier module calculated expression and fold randfold scores. High-scoring candidates were classified as novel miRNA precursors, with their corresponding reads designated as novel miRNAs.
L184-187
Conditions for PCR when constructing sRNA libraries were added:The PCR conditions were set as follows: initial denaturation at 94°C for 3 minutes, followed by 15 cycles of denaturation at 94°C for 15 seconds, annealing at 65°C for 15 seconds, and extension at 72°C for 15 seconds, with a final extension at 72°C for 1 minute and hold at 4°C.
L252-258
Specific information on primer design for qPCR was added: Quantification was performed using the poly(A)-tailing-based RT-qPCR method. This approach employed miRNA-specific forward primers designed against mature sequences and a universal reverse primer (R: GATCGCCCTTCTACGTCGTAT) complementary to the 3'-adapter sequence [42]. The forward primers were designed using the online primer design system from Sangon Biotech (Shanghai, China) (https://store.sangon.com/newPrimerDesign).
Supplementary Figure 2 Sexually dimorphic expression of DEMs validated by RT-qPC Relative expression levels of eight candidate miRNAs in male (M, blue) and female (F, red) gonads (n = 3 biological replicates). Data are presented as mean ± SD (error bars represent standard deviation of technical triplicates). Statistical significance was determined by two-tailed unpaired Student's t-test with Welch's correction (*p < 0.05, **p < 0.01, ***p < 0.001).
L248
The inclusion of the word randomly in describes the process of choosing DEM validation.
U6 was selected as the internal reference gene based on two key considerations: (1) its established use in prior miRNA studies of Scatophagus argus (Li et al., 2022; Leng et al., 2025), and (2) our experimental validation demonstrating excellent amplification efficiency (97.72%) and linearity (R²=0.9957). While we acknowledge that using multiple reference genes can provide more robust normalization, U6 has been widely adopted as a reliable single reference for miRNA quantification in this species.
Li Z, Guo Y, Ndandala CB, Chen H, Huang C, Zhao G, Huang H, Li G, Chen H. Analysis of circRNA and miRNA expression profiles in IGF3-induced ovarian maturation in spotted scat (Scatophagus argus). Front Endocrinol (Lausanne). 2022 Nov 25;13:998207. doi: 10.3389/fendo.2022.998207. PMID: 36506051; PMCID: PMC9732426.
Leng Z, Lin X, Zhai Y, et al. MiR-9-z/zp2, miR-3154-y/zp2, and miR-3533-y/zp3 participated in ovarian development and adaptation to salinity stress in spotted scat (Scatophagus argus)[J]. Aquaculture Reports, 2025, 40: 102653.
- I am not sure I understand the logic behind the presentation of some of the results as well as where some data came from. For instance, you present that a large amount of your data was for RNA sequence >20 nt and it is not clear how this data was either considered or filtered out for later sections of the manuscript. Did you delete it? Seems pretty small for consideration in microRNA research. What does its prevalence say about the overall quality of your dataset? Also, what is going on with the sequence >1000 bp? Wasn't this NovoSeq data? How were the sequences so long? The validation study also should have been more expansive than 6 targets all >4 log(FC). Data analysis include samples down to 2 log FC and error bars on RNA-seq data in Figure 5. The lack of clarity here undermines your other compelling results.
Reply:
We sincerely appreciate the reviewer's insightful comments regarding our miRNA data processing. We have added detailed descriptions of the miRNA sequencing identification process (L199-324). All filtered sequences were confirmed to be non-miRNAs, and their removal helped refine subsequent miRNA identification. The initial mention of sequences >1000 bp in our description was incorrect. Our data were indeed analyzed using NovaSeq. These results describe the proportion of miRNA expression levels in different ranges after TPM normalization. Correction in L 319-324 ”TPM normalization analysis revealed that the majority of miRNAs (37.15%, n=821) exhibited expression levels between 10-100, followed by those in the 0-10 range (34.84%, n=770). miRNAs with expression >1000 accounted for the smallest proportion (5.88%, n=130) (Figure 2A).” For the validation of DEMs, we included one additional DEM from each sex to broaden the log2FC range (>2) among the validated DEMs. The primers for the new DEMs were added to Table 1, and the primer order was adjusted to match the validation data order in Figure 4 (the figure order was rearranged due to the deletion of the original Figure 3). The analysis of error bars has been revised accordingly. For RT-qPCR calculations, the same method as RNA-seq was used, calculating based on the mean expression levels of male and female groups.
- The Joint miRNA/Transcriptome Analysis section would benefit from a bit more detail and some clarity of language as well. First, what is the level of confidence we should be associating with the 462 DEMs associatedw ith 3,340 DEGs and the potential 13,673 DEM-DEG pairs? This process seems rife false positives and seems to involve a fair amount of conjecture as presented. Could you provide some more information here to convince the reader that these are legitimate matches? I also found the whole section on negative and positive correlations of DEM-DEG pairs to be confusing and difficult to follow. As presented, it feels like the authors are saying that both cases hold equal value and either is equally likely to be expected. I would general think negative correlated expression would be more likely than positive. Regardless, it leaves the Discussion feeling full of conjecture. I think the manuscript would be strengthened by setting clear expectations and interpretations of results earily in the manuscript and then adhering to those expectations for downstream data analysis. Otherwise, the discussion ends up feeling a bit meandering. I found the comparative points to be the most interesting and compelling and would have loved to have seen more about that, especially some kind of explanation for why DMRT1 would be microRNA controlled in one organism and now another?
Reply:
The suggestion was adopted. We have added a clearer and more detailed analysis process regarding DEM-DEG interactions in L272-281. In the present study, although potential false positives may exist in miRNA sequencing analysis, gene 3'UTR identification, and DEM-DEG pair prediction, our analytical approach was systematically designed based on: (1) intrinsic miRNA characteristics, (2) structural features of gene 3'UTRs, and (3) binding mechanism properties in DEM-DEG regulatory regions to obtain reliable DEM-DEG predictions. While comprehensive experimental validation of all predicted DEM-DEG pairs would require substantial effort, this study provides a fundamental framework for subsequent investigations focusing on the verification of gonad development-related genes and their regulatory DEMs within the established interaction network. Additionally, the predictive principles of TargetScan have been supplemented (L287-289).
Indeed, the relationship between miRNAs and their target genes is negatively correlated. We introduced this concept in the introduction (L81-82). Furthermore, the subsequent results and discussion (L451-465) also focus on this negative correlation.
In the discussion section, we explore potential reasons for the differential regulation of DMRT1 and other sex development-related genes by miRNAs across different species .L542-559:The observed interspecies differences in miRNA regulation of the male sex-determination gene Dmrt1Y and other gonad development-related genes (Foxl2, Cyp19a1/Cyp19a1a, Gsdf, and Amh) in S. argus may arise from both technical and biological factors. Since the predicted regulatory relationships between these genes and DEMs have not been experimentally validated in either this study or some comparative studies, the observed variations could result from technical differences in miRNA and 3'UTR sequencing analyses (including sequencing platforms, library preparation methods, and bioinformatics pipelines), as well as discrepancies in target prediction algorithms and their parameter settings. When the predicted gene-DEM regulatory pairs exhibit minimal technical artifacts or have been experimentally validated across species, these observed differences may reflect species-specific characteristics in the miRNA regulatory mechanisms of gonad development-related genes from a biological perspective. In cases where highly conserved miRNAs are identified among the compared species, the differential gene-DEM regulatory relationships may result from species-specific variations in the 3'UTRs of conserved genes. Conversely, when miRNAs show low sequence similarity across species, the observed interspecies differences could originate from either the miRNAs themselves, the 3'UTRs of conserved genes, or both exhibiting species-specific features.
More less significant points:
- Some grammatical mistakes with things like italics, use full Genus and species first time then abbreviate afterwards, and other minor errors. Please proofread.
Reply:
Thank you for your suggestion. Grammar and Formatting : Have performed a comprehensive proofreading pass. Ensured proper italics usage and species nomenclature (full genus/species at first mention, abbreviated thereafter). Corrected all minor grammatical errors throughout
- Third paragraph of the intro is pretty generic. Consider revising it significantly.
Reply:
Have significantly revised the third introductory paragraph (now L107-117) to be more focused and specific.
- I'm not sure what "bottlenecks" you are referring to in the paragraph of the intro
Reply:
Bottlenecks: To establish an all-female breeding system in S. argus, the successful production of sex-reversed XX males for crossing with normal XX females is crucial for generating all-female progeny. However, conventional hormonal treatments using androgen 17α-methyltestosterone (MT) and aromatase inhibitor letrozole (Le) fail to induce complete sex reversal in XX individuals [32]. (L122-127)
- Figure 7 legend needs grammer work
Reply:
Revised Figure 7 (originally Figure 6) legend for better clarity and grammar.
- Figure 3 feels unnecessary - would have been more interesting to see something describing miRNA families visually.
Reply:
We sincerely appreciate the reviewer's valuable suggestion regarding Figure 3. After careful consideration, we agree that the figure's content was not essential to our core findings and have therefore removed it from the manuscript. All subsequent figure numbers have been updated accordingly.
Round 2
Reviewer 1 Report
Comments and Suggestions for Authors
Ya-Ling Lei, Kai-Zhi Jiao, Yuan-Qing Huang, Yu Li, Yu-Wei Wu, Gang Shi, Hong-Juan Shi, Hua-Pu Chen, Si-Ping Deng, Guang-Li Li, Dong-Neng Jiang, Wen-Jing Tao
Integrated analysis of differential expression profiles of miRNA and mRNA in gonads of Scatophagus argus provides new insights into the sexually biased gene expression
animals-3551547
The issues and suggestions I made in my previous review have been satisfactorily addressed by the authors. However, in the submitted PDF version of the text there are sentences/paragraphs that are not marked as deleted. The authors have to be careful that no obsolete fragments remain in the final version (e.g. correction of species names).
I have a couple of additional comments, but my opinion is that the manuscript could be accepted after a minor revision.
- A number of sentences seem to have missing references (Is this an issue related to version control?):
+ L100-101 "MiRNAs control gene expression mainly by binding to the 3' untranslated region (UTR) of target genes, resulting in cleavage of the target messenger RNA (mRNA) or inhibition of mRNA translation[]"
+ L174-177 "Unfortunately, S. argus XX individuals can not be reversed to neomales by conventional hormone therapy, possibly because they lack func- tional Dmrt1 (compared to the Dmrt1Y on the Y chromosome, the Dmrt1ΔX on the X chro- mosome is truncated), and other possibilities cannot be ruled out [,]"
+ L184-286 "For sequences that did not match known miRNAs, po- tential precursor sequences were identified using miRDeep2 (v2.0.5) [] through a compre- hensive genome-wide comparison of the sequencing reads"
+ L342-344 "Transcriptome analyses of mRNA expression profiles from the same testis and ovary samples were conducted by our re- search group []"
- L194-197 "After completing body weight and length measurements (Supplementary Table 1), gonadal tissue samples were immediately collected through dissection. Gonadal tissue samples were immediately col- lected through dissection."
This sentence "gonadal tissue samples were immediately collected through dissection" appears twice.
- L326-327 "Detailed sequence information is provided in Table 1.The primer sequences used in this analysis are provided in Table 1"
Combine both sequences.
- L337-338 "For genes lacking 3'UTR annotations, we developed a Python pipeline to attainment these regions, defined as se- quences between stop codons and polyadenylation signals"
This part of the sentence: "... to attainment these regions...", probably should be changed to "... to attain / obtain these regions...".
- L407-408 "Strikingly, while 13 DEMs were exclusively expressed in ovarian tissue, 17 were exclu- sively expressed in testes tissues."
You should support why these results are "striking" with references to other works.
Author Response
The issues and suggestions I made in my previous review have been satisfactorily addressed by the authors. However, in the submitted PDF version of the text there are sentences/paragraphs that are not marked as deleted. The authors have to be careful that no obsolete fragments remain in the final version (e.g. correction of species names).
Reply:
We sincerely appreciate the reviewer's follow-up comments. Regarding the concern about unmarked deletions in the PDF version, we suspect this might be due to compatibility issues with different PDF viewing software or versions. In our previously revised manuscript, all modifications (including species name corrections and other edits) were properly tracked in the review mode (with changes recorded on the right side). To ensure clarity, we have now regenerated the PDF file with all revisions properly reflected. Could the reviewer please verify whether this new version displays correctly? We remain fully committed to addressing any remaining formatting or content issues.
I have a couple of additional comments, but my opinion is that the manuscript could be accepted after a minor revision.
- A number of sentences seem to have missing references (Is this an issue related to version control?):
Reply:
We appreciate the reviewer's feedback. The missing references were likely due to a file version issue. We have double-checked and corrected them in the revised manuscript.
+ L100-101 "MiRNAs control gene expression mainly by binding to the 3' untranslated region (UTR) of target genes, resulting in cleavage of the target messenger RNA (mRNA) or inhibition of mRNA translation[]"
L79-82: MiRNAs primarily regulate gene expression by binding to the 3' untranslated region (UTR) of target mRNAs. In animals, this binding typically leads to translational repression and/or mRNA degradation (via deadenylation) [16-18].
+ L174-177 "Unfortunately, S. argus XX individuals can not be reversed to neomales by conventional hormone therapy, possibly because they lack func- tional Dmrt1 (compared to the Dmrt1Y on the Y chromosome, the Dmrt1ΔX on the X chro- mosome is truncated), and other possibilities cannot be ruled out [,]"
L123-132:To establish an all-female breeding system in S. argus, the successful production of sex-reversed XX males for crossing with normal XX females is crucial for generating all-female progeny. However, conventional hormonal treatments using androgen 17α-methyltestosterone (MT) and aromatase inhibitor letrozole (Le) fail to induce complete sex reversal in XX individuals [32]. This limitation may be attributed to: the absence of functional Dmrt1 (whereas the Y chromosome encodes intact Dmrt1Y, the X chromosome carries a truncated Dmrt1ΔX variant) [33]. The inability to produce viable XX neo-males currently represents a major technical constraint in developing sex control techniques for S. argus aquaculture.
+ L184-286 "For sequences that did not match known miRNAs, po- tential precursor sequences were identified using miRDeep2 (v2.0.5) [] through a compre- hensive genome-wide comparison of the sequencing reads"
L225-226:These extended sequences were then analyzed using miRDeep2 (v2.0.5) [38]
+ L342-344 "Transcriptome analyses of mRNA expression profiles from the same testis and ovary samples were conducted by our re- search group []"
L272-275:To identify target DEGs regulated by DEMs, we performed differential expression analysis on transcriptome data from the same samples (PRJNA906196 and PRJNA1030442) [35], using the same reference genome as the miRNA analysis (GWHAOSK00000000.1; https://bigd.big.ac.cn/gwh).
- L194-197 "After completing body weight and length measurements (Supplementary Table 1), gonadal tissue samples were immediately collected through dissection. Gonadal tissue samples were immediately col- lected through dissection."
This sentence "gonadal tissue samples were immediately collected through dissection" appears twice.
Reply:
We appreciate the reviewer's keen observation. The duplicated sentence ("Gonadal tissue samples...") has been deleted in the revised version.
- L326-327 "Detailed sequence information is provided in Table 1.The primer sequences used in this analysis are provided in Table 1"
Combine both sequences.
Reply:
We appreciate the reviewer’s careful reading. The redundant mention of Table 1 has been corrected in the revised manuscript (now Lines 268–269): "All primers were synthesized by Sangon Biotech Co., Ltd. Detailed sequence information is provided in Table 1."
- L337-338 "For genes lacking 3'UTR annotations, we developed a Python pipeline to attainment these regions, defined as se- quences between stop codons and polyadenylation signals"
This part of the sentence: "... to attainment these regions...", probably should be changed to "... to attain / obtain these regions...".
Reply:
The suggestion was adopted. L279:For genes lacking 3'UTR annotations, we developed a Python pipeline to obtain these regions, defined as sequences between stop codons and polyadenylation signals.
- L407-408 "Strikingly, while 13 DEMs were exclusively expressed in ovarian tissue, 17 were exclu- sively expressed in testes tissues."
You should support why these results are "striking" with references to other works.
Reply:
We thank the reviewer for this constructive suggestion. As recommended, we have modified the text (L339) to:"Among these, 13 DEMs were exclusively expressed in ovarian tissue while 17 were exclusively expressed in testicular tissue."